# Magnetron Sputtering Deposition of High Quality Cs_3_Bi_2_I_9_ Perovskite Thin Films

**DOI:** 10.3390/ma16155276

**Published:** 2023-07-27

**Authors:** Stefano Caporali, Stefano Mauro Martinuzzi, Lapo Gabellini, Nicola Calisi

**Affiliations:** 1Department of Industrial Engineering (DIEF), University of Florence, Via di Santa Marta n. 3, 50139 Florence, Italy; stefano.caporali@unifi.it (S.C.); lapo.gabellini@unifi.it (L.G.); 2National Interuniversity Consortium of Materials Science and Technology (INSTM), Florence Research Unit, Via G. Giusti n. 9, 50121 Florence, Italy; 3Department of Chemistry “Ugo Schiff” (DICUS), University of Florence, Via della Lastruccia n. 3, 50019 Sesto Fiorentino, Italy; stefanomauro.martinuzzi@unifi.it

**Keywords:** perovskite, Cs_3_BI_2_I_9_, lead-free, temperature, magnetron sputtering

## Abstract

Nontoxic all-inorganic perovskites are among the most promising materials for the realization of optoelectronic devices. Here, we present an innovative way to deposit lead-free, totally inorganic Cs_3_Bi_2_I_9_ perovskite from vapor phase. Taking use of a magnetron sputtering system equipped with a radiofrequency working mode power supply and a single target containing the correct ratio of CsI and BiI_3_ salts, it was possible to deposit a Cs_3_Bi_2_I_9_ perovskitic film on silicon and soda-lime glass. The target composition was optimized to obtain a stoichiometric deposition, and the best compromise was found with a mix enriched with 20% *w*/*w* of CsI. Secondly, the effect of post-deposition thermal treatments (150 °C and 300 °C) and of the deposition on a preheat substrate (150 °C) were evaluated by analyzing the chemical composition, the morphology, the crystal structure, and the optical properties. The thermal treatment at 150 °C improved the uniformity of the perovskite film; the one at 300 °C damaged the perovskite deposited. Depositing on a preheated substrate at 150 °C, the obtained film showed a higher crystallinity. An additional thermal treatment at 150 °C on the film deposed on the preheated substrate showed that the crystallinity remains high, and the morphology becomes more uniform.

## 1. Introduction

Thanks to their interesting optoelectronic properties, mixed organic–inorganic lead halide perovskites (e.g., CH_3_NH_3_PbX_3_, where X is a halide element) have been proposed for many applications, ranging from laser emitter [1], sensitive material for photon sensors [2], and photovoltaic cells [3]. However, the presence of the organic part in their structure causes an intrinsic instability due to the volatility of this cations in the presence of moisture [4,5]. To overcome this instability problem, some inorganic cations were proposed for the substitution of methylammonium, such as cesium or rubidium, to form the fully inorganic lead halide perovskites, obtaining an effective improvement of the stability to atmospheric condition [6]. Another discussed point in the use of lead halide perovskites as an absorber layer in solar cells is the presence of lead, which is highly toxic [7]. To overcome this problem, other cations were proposed such as tin, bismuth, magnesium, etc. [8,9,10]. Even though their photovoltaic conversion efficiency (PCE) cannot equalize the performances of lead-based ones, their intrinsically reduced environmental concerns make them suitable to become the base materials of next-generation large scale photovoltaic technologies [11].

Among the elements potentially suitable to substitute lead, bismuth, being a trivalent cation with lone-pair states, has attracted significant interest, not only because the compounds containing bismuth show the possibility to be used as light adsorber in solar cells with a possible maximum conversion efficiency of 9.54% [12], but also thanks to its low water solubility that makes Bi-based perovskites more stable in ambient conditions [13] with respect to tin or magnesium ones. Among the Bi-based compounds, Cs_3_Bi_2_I_9_ seems to be the most suitable for optical application, since it is characterized by a suitable direct bandgap (2.1–2.2 eV) [14,15]. Cs_3_Bi_2_I_9_ was also proposed as a photocatalyst for organic and inorganic pollutant degradation [16,17]. The structure consists of isolated face-shared [B_2_X_9_] dioctahedral as depicted in the scheme of Figure 1.

Such low structural dimensionality (virtually 0D) turns into low electronic mobility which limits the photovoltaic efficiency. Despite these drawbacks for application in photovoltaic cells, the fabrication of highly ordered thin films of Cs_3_Bi_2_I_9_ still remains highly sought. Different synthesis paths were proposed. Single step wet ones consist of dissolving both the precursor salts in a common solvent (dimethylformamide, dimethyl sulfoxide, a mix of them, or in hydrogen iodide) and obtaining the thin films after the deposition by drop-casting or spin-coating and the evaporation of the solvent [18,19]. Variation and optimization of such synthesis approach were proposed [20] as well as thin film deposition via spray coating [21] and the growing the material between two glass slides to obtain a flat deposit [22]. Via wet chemistry, the growth of single crystal was also obtained [23].

Physical vapor deposition (PVD) techniques are largely used in solar cell fabrication and more generally in electronics, thanks to their ability to produce thin and smooth films of virtually all materials on all substrates. For these reasons, PVD techniques were proposed also for the fabrication of perovskite based solar cells [24,25]. Obtaining a smooth and uniform film of perovskite is also essential to obtain an improvement in its time stability [26]. Magnetron sputtering is a PVD technique that uses plasma to produce the etching of the target and the formation of particles that can deposited on the substrate. This technique was successfully used by our group for the deposition of several inorganic perovskites such as CsPbBr_3_ [27] and CsPbCl_3_ [28].

In this work, we proposed the use of magnetron sputtering for the deposition of the fully inorganic lead-free perovskite Cs_3_Bi_2_I_9_. The composition of the target used for the sputtering process was optimized to obtain the deposition of the desired composition. Then, the nature of the obtained films was evaluated, and the presence of the perovskite was confirmed by X-ray photoelectron spectroscopy (XPS), scanning electron microscopy (SEM), X-ray diffraction (XRD), and UV–Vis absorption and reflectance techniques. The effect of annealing treatment and of the deposition on a preheated substrate were also investigated and optimized to achieve the most homogeneous and uniform layer.

## 2. Materials and Methods

### 2.1. Materials

Cesium iodide, 99% pure (purchased from ChemPUR, Prod. N.: 001202, CAS N.: 7789-17-5), and bismuth iodide, 99% pure (purchased from Sigma Aldrich, Prod. N.: 341010, CAS N.: 7787-64-6), as well as acetone and ethanol (technical grade, purchased from Sigma Aldrich, Merck KGaA group, Darmstadt, Germany), used for cleaning the substrates, were used as they were, without any further purification.

Substrates are constituted by soda-lime glass and silicon, both obtained cutting 2 × 2 cm squares, the first ones from microscope slides and the second ones from (100) oriented B-doped silicon single crystal (Sigma Aldrich, Merck KGaA group, Darmstadt, Germany). Before use, they were cleaned by sonication in ethanol and acetone and then dried with a dry air flow.

### 2.2. Instrumentations

XPS consists of an X-ray source, VSW Scientific Instrument Limited model TA10, taking use of Mg Kα radiation at 1253.6 eV, and a hemispherical magnetic analyzer, VSW Scientific Instrument Limited, equipped with a 16-channel detector. The obtained spectra were fitted using the dedicated software CasaXPS, subtracting the background signal using the Shirley’s method [29] and fitting the peaks using a mixed Gaussian–Lorentzian function. Calibrations of the spectra were made by imposing the C 1 s component of adventitious carbon at 284.8 eV [30].

SEM images were carried out with a Hitachi SU3800 SEM (Hitachi High-Tech, Tokyo, Japan), using an acceleration voltage of 15 keV and magnification factor of × 1.00 k and × 5.00 k. Before acquiring the images, samples were metallized using a sputter coater (Polaron Range) equipped with an Au/Pd alloy target.

XRD patterns were acquired in a Bruker model D8 advanced diffractometer working in Bragg–Brentano mode.

The UV–Vis absorption spectra and direct bandgaps were estimated by acquiring the spectra in the UV–Vis region using an Agilent Cary 300 spectrophotometer in transmittance and reflectance mode, using the samples deposed on the glass slide substrates. The spectra were registered in the range 200–800 nm, at a step of 1 nm, counting 0.1 s per step.

Thin films were sputtered using a Korvus HEX magnetron sputtering (Korvus Technology Ltd., Newington, UK). The instrument is equipped with a radio frequency source working at 13.56 MHz and power of 20 W. With a mass flow control, an argon flow of 35 sccm was imposed, corresponding to a dynamic working pressure in the deposition chamber in the order of 10^−3^ mbar. The deposition rate was monitored using a quartz microbalance and set to 0.05 nm/s up to 500 nm deposition thickness. To guarantee a homogeneous deposition, the substrates were fixed on a rotating sample holder set to 10 rpm during the deposition process. The sample holder can also heat up the substrate up to 500 °C. In this work, the depositions were obtained on substrates at room temperature and 150 °C. Some samples received after deposition a thermal treatment at 150 °C and 300 °C for 24 h: in the first case it was carried out in a normal laboratory oven (model ED 115, Binder, Tuttlingen, Germany), and in the second it was carried out in a muffle (CWF 1300, Carbolite Gero Ltd., Verder Scientific GmbH & Co. group, Haan, Germany).

### 2.3. PVD Targets Preparation

Desired amounts of the precursor salts (CsI and BiI_3_) were ground together in an agate mortar to produce a fine and homogeneous powder. The powder was inserted in a steel mold and pressed for 12 h by means of a pneumatic press working at 11.5 MPa and 150 °C. After the sintering process, 5 cm diameter disks were obtained and used as target in the magnetron sputtering. To optimize the sputtered film composition, three targets with different compositions were prepared as reported in Table 1.

### 2.4. Thin Films Preparation

Seven types of 500 nm-thick films were obtained by magnetron sputtering varying target composition, temperature of the substrate, and post-deposition treatment, as summarized in Table 2. The deposits were obtained both on soda-lime glass and silicon. The samples obtained on soda-lime glass were used for XRD analysis, avoiding in such way strong diffraction peaks from the substrate and UV–Vis measurements, while the samples deposited on silicon were used for chemical and morphological investigations. Both the substrates were in the sputtering chamber at the same time to avoid differences between the films. After deposition, the samples were stored in a desiccator to prevent their eventual atmospheric contamination and moisture damage.

Samples were labeled using this code: Excess of CsI in the target _ Deposition temperature _ Thermal treatment temperature

## 3. Results and Discussion

### 3.1. Structural Characterization

XRD spectra were collected to check the successfully perovskite deposition, an example (Sample 20_0_0) depicted in Figure 2.

Two strong reflexes are clearly detectable: the first one is at about 27°, attributable at the (203) reflex of Cs_3_Bi_2_I_9_ (PDF 04-008-8708), confirming the presence of this mineralogical specie. The second one at about 49° (marched with circle in Figure 2), not directly attributable to the perovskite structure, could be attributed to the (211) reflex of CsI. It is possible to suppose that during the deposition the reaction of the precursors to form perovskite happens just partially. Noticeably, only one diffraction peak can be clearly attributed to Cs_3_Bi_2_I_9_, and this peak coexists with a large hump attributable to amorphous phases. Such experimental evidence suggests the presence of highly textured crystalline phases (only one peak clearly detectable) along with an amorphous phase, for which it is well known that it is the most common phase for PVD-grown samples [27].

### 3.2. Crystalline Structure Evaluation

The effect of the thermal treatments on the deposited thin films was monitored by XRD. Results are depicted in Figure 3 on which wide scans spectra are reported together with zooms in of the (203) characteristic perovskite diffraction peak.

Notably, such a deposit evidenced a remarkable variation in function of the thermal treatment carried out on the perovskite film. The intensity of the diffraction peak attributed to cesium iodide substantially decreased after the 150 °C thermal treatment, suggesting that the thermal treatment promotes the formation of perovskite from the precursor salts. Over that, the intensity of the (203) diffraction peak of Cs_3_Bi_2_I_9_ increases as a function of the treatment temperature, while the contribution of the amorphous phase decreased. The diffraction pattern of Sample 20_0_300 (blue curve in Figure 3a) does not show a change in the relative intensity of the two peaks, but it is possible to note a small shift of the left one reflex to about 28° (Figure 3a, in the zoom on the right), confirming the formation of bismuth oxide (reflex (201), PDF 01-083-3010). Figure 3b reports the diffraction pattern of Sample 20_150_0 and Sample 20_150_150 compared to the one of Sample 20_0_0. These patterns shown that no CsI was present in the deposited thin films and the reflex attributed to perovskite appears more intense and sharper, signaling that the obtained films were more crystalline and bigger grains were obtained compared to Sample 20_0_0. Moreover, there were no appreciable differences between the patterns obtained from Sample 20_150_0 and Sample 20_150_150.

### 3.3. Chemical Characterization

Aiming to obtain Cs_3_Bi_2_I_9_ thin films by single step magnetron sputtering technique, different PVD targets were obtained and tested (Table 1). Checks of correct stoichiometry of the deposited thin films were demanded for XPS analysis. Figure 4 depicted the XPS survey spectra of Sample 0_0_0.

The spectrum confirmed the presence of all the desired elements: iodine, cesium, and bismuth, respectively, marked in Figure 4 in yellow, green, and blue dash lines and labeled, respectively, with a, b, and c. The fitting of these peaks (Table 3) revealed that the stoichiometry of Sample 0_0_0 is different from the one of the target. Significant cesium deficiency and excess of bismuth are detectable as a consequence of the different sputtering yield of elements [31]. It was also possible to recognize in the spectrum the peaks relative to the 1 s transitions of carbon and oxygen, both marked with a red line and labeled with the letter d on the spectrum. These two elements were attributed firstly to adventitious carbon, a pollutant ever-present in XPS analysis, due to the superficial adsorption of carbonaceous substances from the air. Oxygen could be attributed also to the reaction of the excess of bismuth with the atmospheric oxygen.

To obtain a stoichiometric deposit, two targets with an increased amount of CsI (15% *w*/*w*, Sample 15_0_0, and 20% *w*/*w*, Sample 20_0_0) were prepared. The compositions of the targets are summarized in Table 1 both for molar and weight ratio. The elemental composition of samples was evaluated by XPS analysis. The high-resolution spectra in the regions of the 3d transition of cesium (from 745 eV to 715 eV), 3d transition of iodine (from 640 eV to 610 eV), and 4f transition of bismuth (from 170 eV and 155 eV) for the three samples are reported in Figure 5a, Figure 5b, and Figure 5c respectively.

Data were normalized by dividing for the maximum value measured in the three regions of the sample after the background subtraction. In such way, the relative intensities of the spectra are comparable between samples in the same region but not between different elements of the same sample, because it does not consider the atomic sensitivity factors. It was possible to note that the relative intensities of the reported spectra changed by changing the amount of cesium in the target. In particular, the amount of cesium increased, causing the decrease of the intensities of the other elements. It is also noted that there is a small shift of the binding energy toward lower energy of the cesium, bismuth, and iodine peaks. This is probably related to the formation of the perovskite structure, for which the chemical environment of the constituted elements differs from the reactants, according with previously reported value of binding energy [32,33].

The data obtained by the fitting of the spectra reported in Figure 5 are summarized in Table 3 together with the expected values for the three elements and Cs/Bi ratio, the value used to optimize the composition of the target.

By increasing the amount of cesium in the target, the Cs/Bi ratio increased until reaching a value close to the expected one (1.55 for the film deposited with the target +20% *w*/*w* CsI). On the other hand, the amount of iodine in the film also increased but remained lower than the expected (less than 50% compared to an expectation of 64%). In any case, the closest value to stoichiometric one was obtained with the target +20% *w*/*w* CsI, and then this target was used to produce the other samples.

### 3.4. Surface Elemental Composition Analysis

The elemental composition of the surface of the samples was determined by XPS, and the results are summarized in Table 4. For a correct interpretation of these results, it is important to remember that the probing depth of this technique is about 2 × 10^−9^ m (2 nm) due to the low-electron mean free path in solid materials. To determine the amount of cesium, bismuth, and iodine in the samples, the peaks were fitted, and tabulated sensitivity factors [34] were applied. The peak relative to the 1 s transition of oxygen was fitted with two components: the first one at about 531.0 eV relative to oxygen in organic matrix, attributable to adventitious carbon, and the second one at about 529.6 eV, relative to oxygen in metal oxides. This second component was used to estimate the oxidized state of bismuth and cesium in the deposited films, and the results are displayed in Table 4. In our experimental conditions, it is not possible to discriminate cesium and bismuth oxides.

XPS results confirm the oxidation of the sample after the 300 °C thermal treatment since in Sample 20_0_300 the amount of oxygen increases until about 40%, together with the content of bismuth. Considering the energy shift of the 4f bismuth core peak transition, the degradation of the perovskite can be assessed, involving the formation of bismuth oxide which migrates on the surface of the sample returning the high observed value (see Table 4). Notably, after the thermal treatment at 150 °C (Sample 20_0_150 and Sample 20_150_150) the bismuth amount, as well as the oxygen, decreases. This phenomenon can be explained by supposing that after the deposition process a thin layer of bismuth oxide is formed on the surface of the samples, as a passivation layer. After the thermal treatment, a reorganization of the films is produced, leading to the homogenization of the deposited layer.

### 3.5. Thermal Treatment: Optical Characterization

To obtain a complete optical characterization, both absorbance and reflectance spectra were acquired on samples. The absorbance spectra are depicted in Figure 6a. To determinate the bandgaps, the UV–Vis spectra were acquired in total reflectance mode, fitting the data by means of McLean analysis at the absorption edge [35] using an n equal to 2, corresponding to an allowed direct bandgap. In particular, the bandgap values were estimated by the extrapolation of the linear trend in the Tauc plot (Figure 6b) [36,37], and the results were reported in Table 4.

UV–Vis spectra show a good transmittance of the light over the 500 nm. Below this, wavelength samples (except 20_0_300) show different absorption edges due to the interaction of perovskite with the light. After these absorption edges, the rapid increasing of the absorbance can be attributed to the glass absorption (substrates). Sample 20_0_0, Sample 20_150_0, and Sample 20_150_150 present an absorption in the range of 500–550 nm, corresponding to an energy of about 2.25–2.50 eV, comparable to the Cs_3_Bi_2_I_9_ bandgap value. Sample 20_0_150 has the absorption edge at a lower value, corresponding to a higher energy. 

The measured bandgaps for Sample 20_0_0, which was deposited at room temperature and without successive thermal treatment, resulted slightly higher than the expected one (2.26 eV). That is reasonably due to the low amount of iodine content which leads to an incorrect stoichiometry. In these samples, the thermal treatments promote a bandgap increase of 2.45 eV after heating at 150 °C and 2.61 eV after heating at 300 °C. Vice versa, the samples deposited at 150 °C (Sample 20_150_0 and Sample 20_150_150) do not demonstrate bandgap variations indicating, overall, superior structural stability as already evidenced by XRD patterns (Figure 3).

### 3.6. Thermal Treatment: Morphological and Chemical Effect

The evolution of the deposited films as a function of thermal treatment was monitored by XPS and SEM. The high-resolution spectra in the region of 3d transition of cesium (from 745 eV to 715 eV), 3d transition of iodine (from 640 eV to 610 eV), and 4f transition of bismuth (from 170 eV and 155 eV) for the three samples are depicted in Figure 7a–c. From a chemical point of view, cesium and iodine do not present any modification as evidenced by the absence of chemical shift in the peaks depicted in Figure 7a,b. Vice versa, the peak of bismuth shifts toward lower binding energy after the thermal treatment at 300 °C. That is reasonably attributable to a chemical evolution with respect to the initial state. According to the increasing of the oxygen amount (Table 4), BiO^+^ species can be inferred. To summarize, the thermal treatment at 150 °C does not cause any degradation of the film; instead, a thermal treatment at 300 °C adulterates the perovskite film causing the partial oxidation of bismuth.

SEM images of Sample 20_0_0, Sample 20_0_150, and Sample 20_0_300 were reported on Figure 7 (d1–f2) at two different magnifications: ×1.00 k and ×5.00 k.

Sample 20_0_0 presents a uniform film covered of smalls crystalline structures with an average diameter smaller than 1 µm. For all the samples, the heating treatment stimulates the growing of the crystalline structures, up to an average diameter of 2/3 µm, and the decrease of their number, which consequently leads to increase the roughness. That is compatible with the Stranski–Krastanov growing mode: mixed island and layer [38]. The absence of the signal from the substrate in the XPS analysis confirms the complete covering of the silicon from the perovskite films.

### 3.7. Deposition Temperature Effect

The effect of deposition on preheated substrate was evaluated in Sample 20_150_0 and Sample 20_150_150 (see Table 2). In Figure 8, data collected on these two samples are compared.

The high-resolution XPS spectra (Figure 8a–c) confirmed that preheating the substrate, as well as the post-deposition treatment at 150 °C, does not damage the perovskite films. SEM images depicted in Figure 8 evidenced a dramatical morphological change: the surface of the Sample 20_150_0 was constituted by well-distributed small crystals with an average diameter of less than 500 nm. In such case, the more representative growing mode is a layer-by-layer one, where a uniform film is formed, leading to classical columnar growth [39]. Sample 20_150_150 shows the presence of the same small crystals, but the surface appears smoother and more compact. 

## 4. Conclusions

Here, we demonstrated the possibility to use the magnetron sputtering technique for the deposition of Cs_3_Bi_2_I_9_ lead-free fully inorganic perovskite. The deposition yield of bismuth was found to be higher than the cesium one, leading to a bismuth enriched deposit. To achieve a closely stoichiometry film, the amount of cesium iodide in the target was increased up to 20% *w*/*w*, obtaining the correct Bi to Cs ratio. The effect of a post-deposition annealing was investigated evidencing that open-air 150 °C treatment increases the morphological uniformity and the crystallinity, without degrading the perovskite film. After deposition, the samples show a tiny surface layer of bismuth oxide that disappears after the post-deposition thermal treatment. Instead, a treatment at 300 °C causes the degradation of the perovskite film, with the formation of bismuth oxide and cesium iodide. The deposition carried out on the pre-heated substrates at 150 °C returned a more uniform film, with higher crystallinity. In these samples, post-deposition annealing at 150 °C promotes an ulterior increasing of the uniformity, composition, and crystallinity of the film.

In conclusion, this investigation opens a new promising way to obtain Cs_3_Bi_2_I_9_ thin films by magnetron sputtering. The quality of the films can be enhanced by depositing the material onto preheated substrates and following an open-air annealing.

## Figures and Tables

**Figure 1 materials-16-05276-f001:**
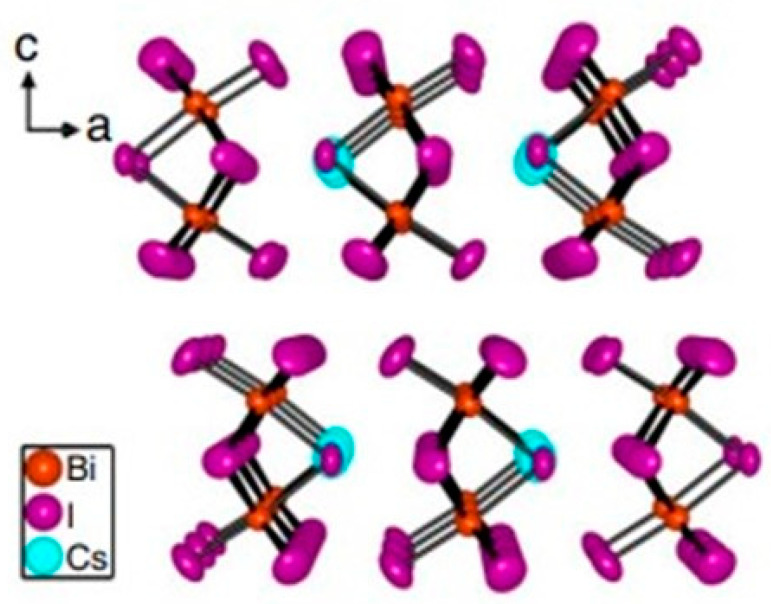
Scheme of the crystal structure of Cs_3_Bi_2_I_9_. Modified form [10].

**Figure 2 materials-16-05276-f002:**
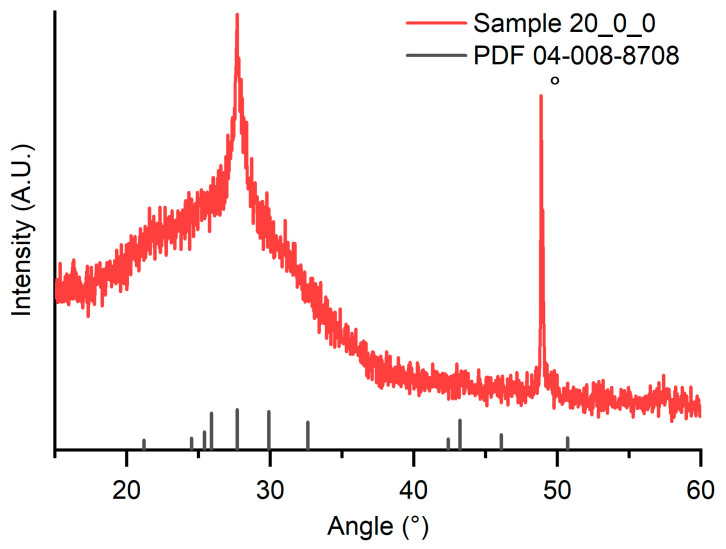
XRD diffraction pattern of Sample 20_0_0 (unreacted CsI reflex is marched with circle).

**Figure 3 materials-16-05276-f003:**
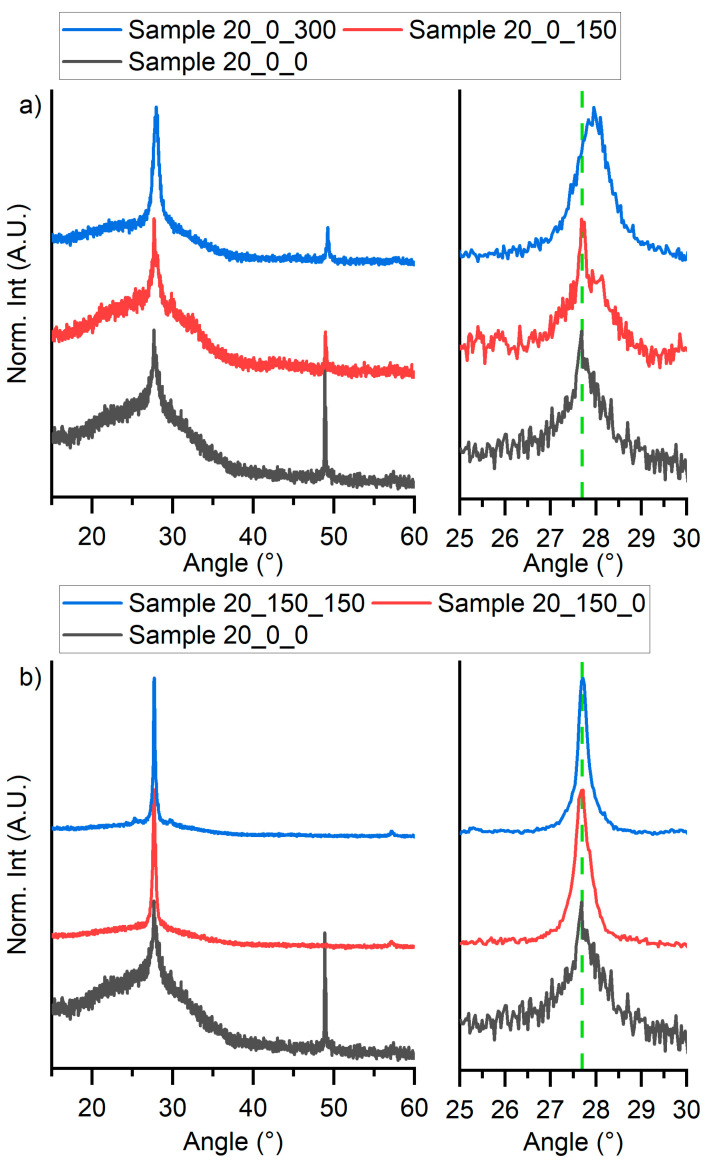
XRD pattern of (**a**) Sample 20_0_0, Sample 20_0_150, and Sample 20_0_300 on the left and the zoom of the principal peak on the right and (**b**) Sample 20_0_0, Sample 20_150_0, and Sample 20_150_150 on the left and the zoom of the principal peak on the right.

**Figure 4 materials-16-05276-f004:**
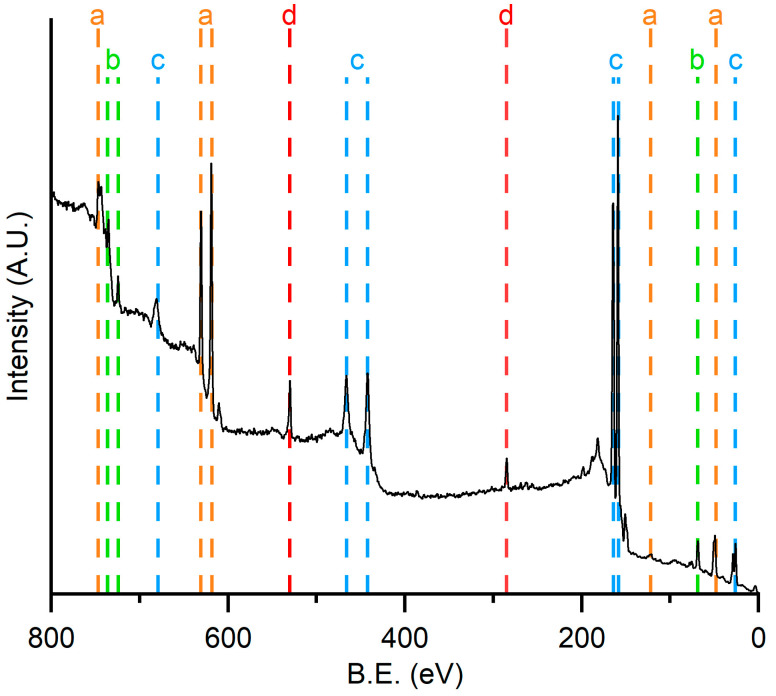
XPS survey spectrum of Sample 0_0_0. Peak marks: yellow for iodine (**a**), green for cesium (**b**), blue for bismuth (**c**), and red (**d**) for oxygen (531 eV) and carbon (285 eV).

**Figure 5 materials-16-05276-f005:**
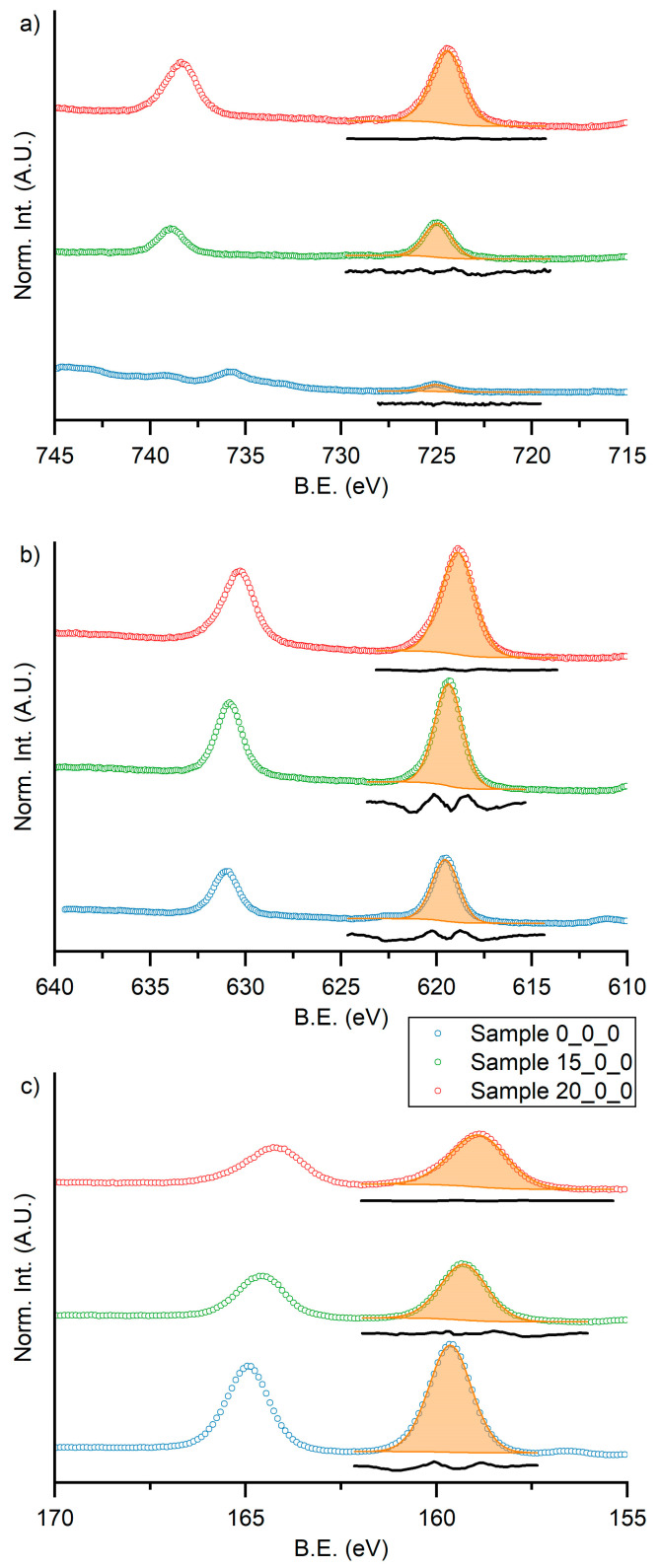
XPS high-definition spectra in the region of (**a**) 3d transition of cesium, (**b**) 3d transition of iodine, and (**c**) 4f transition of bismuth for Sample 0_0_0 (blue dots, stoichiometric target), Sample 15_0_0 (green dots, +15% *w*/*w* CsI target), and Sample 20_0_0 (red dots, +20% *w*/*w* CsI target). The orange areas are the fitted component for each principal peaks, and the black lines are the resulting residuals.

**Figure 6 materials-16-05276-f006:**
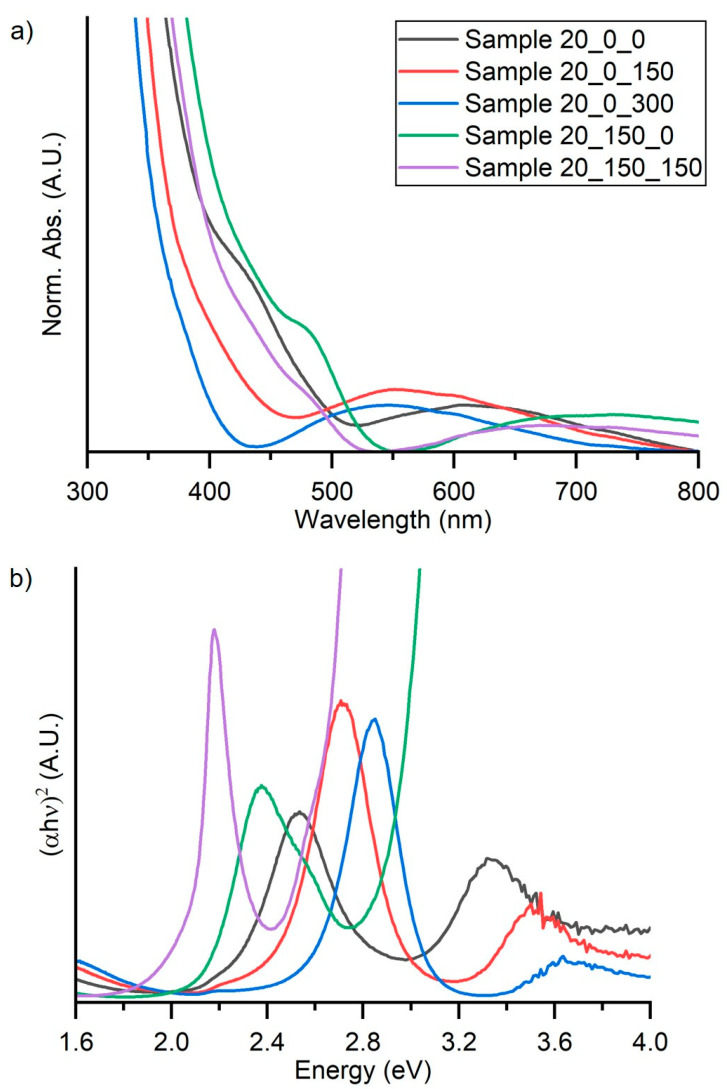
(**a**) UV–Vis spectra and (**b**) Tauc plots of the perovskite thin films obtained from total reflectance spectra.

**Figure 7 materials-16-05276-f007:**
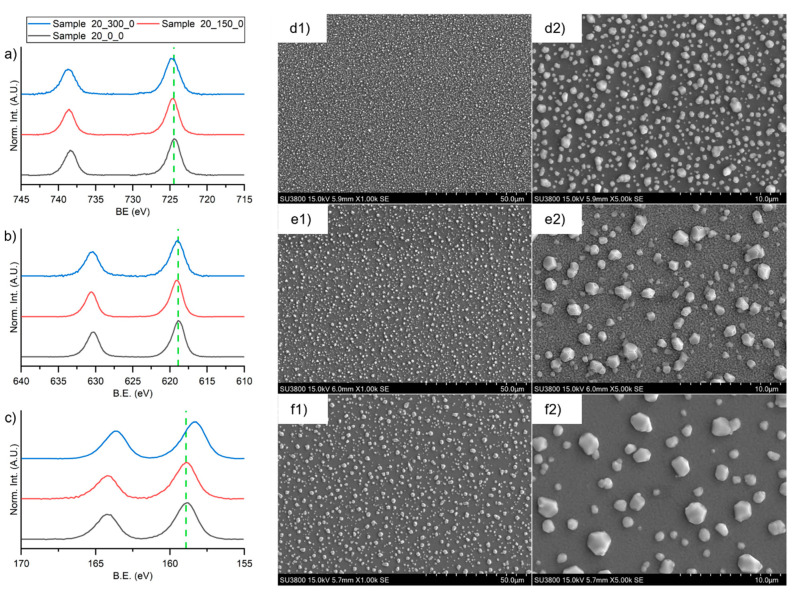
XPS high-definition spectra in the region of (**a**) 3d transition of cesium, (**b**) 3d transition of iodine, and (**c**) 4f transition of bismuth for Sample 20_0_0 (black lines), Sample 20_0_150 (thermally treated at 150 °C after deposition, red lines), and Sample 20_0_300 (thermally treated at 150 °C after deposition, blue lines). SEM images of (**d1**,**d2**) Sample 20_0_0, (**e1**,**e2**) Sample 20_0_150, and (**f1**,**f2**) Sample 20_0_300.

**Figure 8 materials-16-05276-f008:**
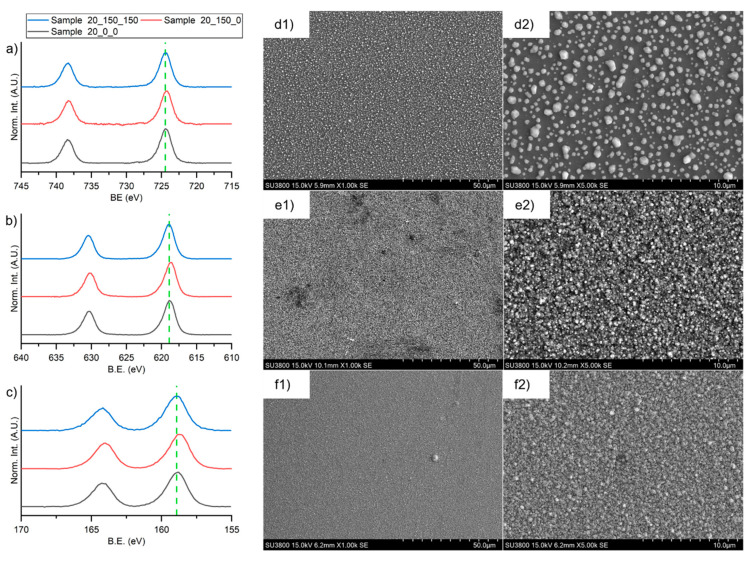
XPS high-definition spectra in the region of (**a**) 3d transition of cesium, (**b**) 3d transition of iodine, and (**c**) 4f transition of bismuth for Sample 20_0_0 (black lines), Sample 20_150_0 (red lines, deposited at 150 °C), and Sample 20_150_150 (blue lines, deposited at 150 °C and thermally treated at 150 °C after deposition). SEM images of Sample 20_0_0 (magnification (**d1**) × 1.00 k and (**d2**) × 5.00 k), Sample 20_150_0 (magnification (**e1**) × 1.00 k and (**e2**) × 5.00 k), and Sample 20_150_150 (magnification (**f1**) × 1.00 k and (**f2**) × 5.00 k).

**Table 1 materials-16-05276-t001:** Target composition.

Target	CsI Quantity		BiI_3_ Quantity	
	Molar (mol)	Weight (g)	Molar (mol)	Weight (g)
Stoichiometric	0.030	7.8	0.020	11.8
+15% *w*/*w* CsI	0.035	9.1	0.020	11.8
+ % *w*/*w* CsI	0.036	9.4	0.020	11.8

**Table 2 materials-16-05276-t002:** Samples description.

Sample ID	Target	Deposition Temperature	Thermal Treatment	Thickness
Sample 0_0_0	Stoichiometric	RT	None	500 nm
Sample 15_0_0	+15% *w*/*w* CsI	RT	None	500 nm
Sample 20_0_0	+20% *w*/*w* CsI	RT	None	500 nm
Sample 20_0_150	+20% *w*/*w* CsI	RT	150 °C, 24 h	500 nm
Sample 20_0_300	+20% *w*/*w* CsI	RT	300 °C, 24 h	500 nm
Sample 20_150_0	+20% *w*/*w* CsI	150 °C	None	500 nm
Sample 20_150_150	+20% *w*/*w* CsI	150 °C	150 °C, 24 h	500 nm

**Table 3 materials-16-05276-t003:** Elemental composition of Sample 0_0_0, Sample 15_0_0, and Sample 20_0_0, obtained fitting the XPS spectra and compared to the expected one.

Element	Elemental Composition (Atomic Percentage)			
	Expected	Stoichiometric	+15% *w*/*w* CsI	+20% *w*/*w* CsI
		Sample 0_0_0	Sample 15_0_0	Sample 20_0_0
Cs	22%	4%	16%	31%
Bi	14%	58%	29%	20%
I	64%	38%	55%	49%
Cs/Bi	1.50	0.08	0.58	1.55

**Table 4 materials-16-05276-t004:** Surface chemical composition and bandgap values of the perovskite thin films.

Sample	Relative Perovskite Composition (Atomic %)			Cs/Bi	O-Metal	Bandgap
	Cs	Bi	I			
Sample 20_0_0	31%	20%	49%	1.55	18%	2.26 eV
Sample 20_0_150	31%	12%	57%	2.55	4%	2.43 eV
Sample 20_0_300	25%	53%	22%	0.46	39%	2.61 eV
Sample 20_150_0	21%	36%	43%	0.57	29%	2.15 eV
Sample 20_150_150	28%	14%	58%	2.06	6%	2.10 eV
Expected	22%	14%	64%	1.50	0%	2.1/2.2 eV [15,16]

## Data Availability

The data presented in this study are available on request from the corresponding author.

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
