# Peer review of "Magnetron Sputtering Deposition of High Quality Cs3Bi2I9 Perovskite Thin Films"

_materials, 2023, doi:10.3390/ma16155276_

Round 1

Reviewer 1 Report

This report proposed the fabrication of Cs3Bi2I9 perovskite films by magnetron sputtering. The authors fabricated their own targets to produce the precursor of the film. Also, they explore some deposit conditions to obtain the perovskites' best properties. The material proposed represent one of the most high interests for solar cell, sensors, transistors, and, recently, photocatalysis; thus, it is important to propose and explore new methods to fabricate devices of this material. Therefore, I recommend its publication after the authors make the following minor corrections:

1. Figure 1. Include the JCPDS reference of the perovskite. Also, identified each reflection since no legend was included about the * or o symbols.

2. Figure 3. Improve the quality of the graphs, please revise the comment 1.

3. Page 6, lines 195-196. Why the authors mentioned XPS results if there is no results (at leat until this part).

4. Figure 4. Identify all the contributions with letters.

5. Figure 5. It is necessary to deconvolute all the signals.

6. Section 3.5. Are you sure that these samples exhibited a direct band gap?

7. If it is possible include AFM results since the SEM images are not clear.

8. Include recent references about this material:

https://doi.org/10.3390/catal12111410

https://doi.org/10.3390/nano10040763

Moderate English changes should be included.

Reviewer 2 Report

In this work, the authors proposed the use of magnetron sputtering PVD for the deposition of the fully inorganic lead-free perovskite Cs3Bi2I9.The nature of the obtained films was evaluated, and the presence of the perovskite was confirmed by X-ray photoelectron spectroscopy (XPS), Scanning electron microscopy (SEM) and X-ray diffraction (XRD) techniques.

Major revision is needed.

1.     All characterizations of the perovskite are only morphology analysis. Other characterizations like absorption/ PL should be include.

2.     The perovskite stability should be included. (moisture?... )

3.     The authors may add some theoretical analysis in the main text.

Besides, Extensive editing of English language is required.

The figure quality is very poor, the authors must reorganize their figures and modify their manuscript. It is now like a temporary draft.

e.g. P3 line 91 “Substrates are constituted by soda-lime glass and silicon, both were obtained cutting 91 2 x 2 cm square…” grammar mistakes.

e.g. abbreviation problem, line 80 and line 96 “X-ray photoelectron spectroscopy (XPS)”

Extensive editing of English language is required

Reviewer 3 Report

In this work, the authors systematically studied the vapor-deposited Cs3Bi2I9 films. The influence of precursor ratio, thermal treatment and substrate temperature are studied in detail.

  The samples were characterized with XRD, XPS, SEM and analyzed clearly. The manuscript is accepted for publication on the journal of Materials after addressing the following issue.

1.      The English should be polished carefully.

The language should be polished carefully.

The first sentance of abstract "Nontoxic all-inorganic perovskites are among of the most promising materials for the realization of optoelectronic devices" should be "Nontoxic all-inorganic perovskites are among the most promising materials for the realization of optoelectronic devices."

Round 2

Reviewer 2 Report

accepted.

Author Response

We would like to say thank you to the reviewer for accepting the manuscript for the publication.